

# Simulating future salinity dynamics in a coastal marshland under different climate scenarios

Julius Eberhard[1], N. Loes M.B. van Schaik[2], Anett Schibalski[3], and Thomas Gräff[4]

[1]Universität Potsdam, Mathematisch-Naturwissenschaftliche Fakultät, Institut für Physik und Astronomie
[2]Technische Universität Berlin, Fakultät VI Planen Bauen Umwelt, Institut für Ökologie
[3]TU Braunschweig, Fakultät Architektur, Bauingenieurwesen und Umweltwissenschaften, Institut für Geoökologie
[4]Umweltbundesamt, IV 2.1

**Correspondence:** Julius Eberhard (e.julius@web.de)

**Abstract.** Salinization is a well-known problem in agricultural areas worldwide. For the last 20–30 years, rising salinity in the upper, unconfined aquifer has been observed in the Freepsumer Meer, a deep grassland area near the German North Sea coast. In order to investigate long-term development of soil salinity and water balance, the one-dimensional SWAP model was set up and calibrated for a soil column in the area, simulating water and salt balance at discrete depths for 1961–2099. The

model setup involved a deep aquifer as the only source of salt through upward seepage since other sources were negligible. In the vertical salt transport equation, only dispersion and advection were included. Six different regional outputs of statistical downscaling methods (WETTREG, XDS), based on simulations of different GCMs (ECHAM5, ECHAM6, IPSL-CM5) driven by greenhouse gas emission scenarios (SRES-A2, SRES-B1) and concentration pathways (RCP45, RCP85), were used as scenarios. These comprised different rates of increasing surface temperature and essentially different trends in seasonal rainfall.

The results of the model runs exhibit opposing salinity trends for topsoil and deeper layers: While the projections of some scenarios entail decreasing salinities near the soil surface, most of them project a rise in subsoil salinity with strongest trends of up to $+0.9 \, \mathrm{mg \, cm^{-3} \, (100 \, a)^{-1}}$ at $-65 \, \mathrm{cm}$. The results suggest that topsoil salinity trends are affected by the magnitude of winter rainfall trends while high subsoil salinity trends correspond to low winter rainfall and high summer temperature. Absolute salinity is high in scenarios of high-temperature and low-rainfall summers. How these projected trends affect the

vegetation and thereby future land use will depend on the future management of groundwater levels in the area.

## 1   Introduction

Since decades, soil salinization has been discussed intensely as global threat to agricultural production (Fedoroff et al., 2010; Maas and Hoffman, 1977; McWilliam, 1986). Yields decrease under saline conditions when high-yield crops are salt-intolerant (Maas and Grattan, 1999), and halophytes used as fodder (Masters et al., 2007) as well as saline drinking water may lead to

lower dairy production (Solomon et al., 1995). Great efforts are made to counteract the negative effects of salinization, e. g. by breeding salt-tolerant crop and fodder species (Flowers, 2004; Glenn et al., 1999; Panta et al., 2014; Rozema and Flowers, 2008; Yamaguchi and Blumwald, 2005; Yensen, 2006).





In the past, research focused mainly on management options for salinization (Ghassemi et al., 1995) caused by irrigation in arid regions (Pitman and Läuchli, 2002; Rengasamy, 2006), which is amplified by climate change (Yeo, 1999), the so-called secondary salinization. Issues involving saline irrigation water in other regions have also been addressed more recently (e. g.
Pauw et al., 2015). Primary salinization on the other hand is caused by sea spray, floodings (e. g. Violette et al., 2009), and sea water intrusion (Werner et al., 2013) and is thus confined to coastal areas around the globe (e. g. Bangladesh: Haque, 2006; Korea: Kim et al., 2003). While primary salinization may currently not be as problematic as secondary salinization, rising sea levels will aggravate the issue as shown by recent changes in coastal vegetation (e. g. Williams et al., 1999). Coastal marshlands in central Europe are examples of intensively used agricultural landscapes threatened by rising sea levels and climate changes
(Bakker et al., 1993). Ground and surface water in these regions can be affected by primary salinization in several ways. Sea water intrusion into deep aquifers may occur because marshland sediments are typically horizontally layered and often form well-connected aquifers (Streif, 1990). In the long term, water from deep aquifers can seep upward to the soil through half-confining layers (aquitards; de Louw et al., 2010). In the short term, greater effects can result from small and local defects in the aquitard (boils; de Louw et al., 2010), and filled up former tideways which cut through confining layers (paleochannels; Weerts,
1996). Storm surges possibly affect the salinity of coastal marshlands in cases of dike failures, as floods can spread quickly over the flat surface (Miegel et al., 2016). Additionally, sea spray can contribute to the salinization of surface water (Stuyfzand and Stuurman, 2008). Apart from salt import, the salt concentration in a soil increases through evaporation and plant transpiration, whereas it decreases through precipitation and subsequent infiltration (de Louw et al., 2013). Water management may further alter the soil salinity through pumping. Thus, the water balance additionally influences the temporal development of the salinity
in a soil profile. Under climate change different water balance components are expected to be subject to long-term changes (Jiménez Cisneros et al., 2014) and therefore may cause long-term shifts in the salinity of coastal areas.

As Kliesch et al. (2016) pointed out, quantitative investigations of the processes involved in the salinization of coastal aquifers in many regions are just at the beginning. A study in the Netherlands showed the possible influence of rising temperatures, changing rainfall patterns, and sea-level rise on coastal salinity (Oude Essink et al., 2010). The authors concluded that
rising groundwater salinities should be expected as a result of climate change, with a particularly strong effect on subsiding areas such as polders. They employed a large scale model, which allowed for general answers to regional managing questions at the Dutch coast. Other recent studies in Italy and Germany used small or medium scale models on mostly sandy aquifers (Colombani et al., 2015; Kliesch et al., 2016) and also predicted rising surface salinities. Regarding small-scale processes, salinization through boils in coastal marshlands was investigated by de Louw et al. (2010). However, boils are rarely observed and
other small-scale processes such as rainfall infiltration, evapotranspiration, and upward seepage through locally thin aquitards may dominate the salt balance. Local models with a high spatial resolution can be helpful in the detailed understanding of these processes (e. g., Delsman et al., 2017).

The collaborative research project COMTESS (Sustainable coastal land management: Trade-offs in ecosystem services) investigates the impact of climate change, sea level rise, and different land management options on hydrological conditions, the distribution of coastal vegetation, and ultimately ecosystem service provision of coastal areas at the Baltic and North Sea coast (Karrasch et al., 2017). In this study we evaluated the influence of climate change on the salinity of the Freepsumer



Meer, employing the one-dimensional SWAP model (van Dam, 2000) for a site within the area. This region is a low-lying marshland and subject to salinization despite a clayey soil and a peaty aquitard. SWAP allows for a specific parametrization of most processes which are relevant for water and solute transport in the soil column and is currently widely used in studies

concerning, e. g., evapotranspiration (e. g. Bartholomeus et al., 2015; Minacapilli et al., 2009), surface water management (e. g. Schipper et al., 2015), plant growth (e. g. Bonfante et al., 2017), and salinization (e. g. Kumar et al., 2015). In our study, we particularly used the water balance and salinity output of the model for a quantitative analysis of projected salinity dynamics. We focused on salt input from deep aquifers and long-term changes at the upper boundary through climate change.

## 2   METHODOLOGY

### 2.1   Study area

The Freepsumer Meer is a former lake in the Krummhörn municipality, north of Emden in northwestern Germany, about 8 km off the coast of the North Sea (Fig. 1a). It contains the lowest measured point of the region, which is 2.3 m below sea level. Today it is used mainly as grassland. The area consists of small lots divided by numerous drain ditches. No point of the surface has a greater distance than about 100 m to the next ditch. Thus the soil water balance is largely influenced by the surface

drainage levels, which in turn are kept almost constant by the nearby pumping station (Fig. 1b). No subsurface drainage occurs in the area.

Upward seepage of deep groundwater is not directly apparent from the geological configuration of the region. The thin silty upper layer, which we consider here as the unconfined aquifer, lies on top of an up to 6 m thick layer of clay, followed by the sandy lower aquifer (cf. Fig. 1c). Typical for the site are layers of strongly mineralized Holocene peat between the clay and

the sand. This basal peat is found beneath the Freepsumer Meer but is interrupted in the eastern part (Wildvang, 1938). Both clay and peat layers form an aquitard with a typically low hydraulic conductivity (Verry et al., 2011). Nonetheless, salinization of the Freepsumer Meer has been observed from measurements of the surface water quality starting in 1985. The marshland of the old lake system is strongly compacted, compared to the surrounding younger marshland. As a result, the area is subject to subsidence (Streif, 1990) reducing the load on lower aquifers and thus increasing the vertical pressure gradient, which can

drive upward seepage.

We tried to account for the observed situation by assuming an aquitard between the sand and the unconfined aquifer. We allowed for upward seepage through the aquitard and estimated its effective hydraulic resistance through calibration.

### 2.2   Model

The SWAP (Soil, Water, Atmosphere, and Plant) model was developed at Wageningen University as a tool for simulating

processes regarding water, heat, and solute transport in the soil-water-atmosphere-plant environment, mainly in the unsaturated zone of soils (van Dam, 2000). SWAP numerically solves the Richards equation (Richards, 1931), which describes mass and energy conservative, unsaturated flow of water in a vertical soil column.





Processes that affect the water balance and are included in SWAP are: water movement between soil particles (infiltration, percolation, and capillary rise), evapotranspiration, rainfall interception, surface runoff, bottom flux, plant uptake, drainage, and macropore exchange. Soil hydraulic characteristics are described with the Mualem–van Genuchten equation (Mualem, 1976; van Genuchten, 1980), which relates the hydraulic conductivity $k$ (in units of $\mathrm{cm\,d^{-1}}$) of a soil layer to its water retention curve. The vertical salt transport equation combines the processes of diffusion, dispersion, advection, and salt uptake by plant roots:

$$\frac{\partial c}{\partial t} = \frac{\partial}{\partial z}\left[\Theta_{\mathrm{s}}(D_{\mathrm{dif}} + qL_{\mathrm{dis}})\frac{\partial c}{\partial z}\right] - \frac{\partial}{\partial z}(qc) - S_{\mathrm{root}}, \tag{1}$$

where $c$ is the solute concentration ($\mathrm{mg\,cm^{-3}}$), $t$ the time (d), $z$ the elevation (cm), $\Theta_{\mathrm{s}}$ the saturated water content or porosity ($\mathrm{cm^3\,cm^{-3}}$), $D_{\mathrm{dif}}$ the vertical diffusion coefficient ($\mathrm{cm^2\,d^{-1}}$), $q$ the vertical Darcy velocity ($\mathrm{cm\,d^{-1}}$), $L_{\mathrm{dis}}$ the vertical dispersion length (cm), and $S_{\mathrm{root}}$ the salt uptake rate of plant roots ($\mathrm{mg\,cm^{-3}\,d^{-1}}$).

We simulated daily water contents, pressure heads, and salt concentrations at discrete depths of one-dimensional soil columns as well as water balance components including evapotranspiration and bottom fluxes.

## 2.3 Input data and model parameters

The geometry of the model domain was defined by the vertical extent of the soil columns, the horizontal distances to drain ditches, and their shape. Measurements of soil physical properties were available for depths up to $-80$ cm relative to the surface. Since the groundwater level dropped below $-80$ cm in some model runs, we needed to specify a model column with additional 20 cm of depth for which we assumed the physical properties to be equal to those at $-80$ cm. The distances and shapes of the ditches were recorded during field measurements for this study.

For driving the model at the upper boundary, we obtained daily data on temperature, wind, rainfall, and relative humidity for the Emden weather station (location in Fig. 1a) from the DWD (Deutscher Wetterdienst). Daily global radiation data was provided by meteo-dynamics.de. We used daily measurements of pressure heads in the confined aquifer from a nearby well (Fig. 1a) as the bottom boundary condition for upward groundwater seepage. Any additional salt import through storm surges and sea spray was ignored in the boundary conditions.

The water board I. Entwässerungsverband Emden provided data of the managed mean surface water level. Since the area is used as grassland, we specified the crop schedule as a fixed cycle of one year length with constant plant parameters.

Soil and plant parameter values were either taken from the literature – including those suggested in the SWAP manual (Kroes et al., 2008) –, modeled by pedotransfer functions (PTFs), or directly or indirectly measured. Data of geodetic height, grain size distribution, soil organic matter, soil layer thickness, and bulk density were available from measurements for previous studies (Witte and Giani, 2016). Since SWAP was originally set up and tested in the Netherlands (Kroes et al., 2000) with similar environmental conditions to our study area, we adopted default parameter values where we had no additional information.

We derived the saturated water content $\Theta_{\mathrm{s}}$ ($\mathrm{cm^3\,cm^{-3}}$, Table 1) from on-site measurements (Table A2) following Waller and Harrison (1986) and Rühlmann et al. (2005). Recorded grain size distribution, organic carbon content, and bulk density (Table A2) were used to estimate the other Mualem–van Genuchten parameters $\lambda$ (no unit), $\alpha$ ($\mathrm{cm^{-1}}$), $n$ (no unit), and residual





water content $\Theta_r$ (cm$^3$ cm$^{-3}$) with three PTFs. Where $\Theta_r$ was 0 according to the PTF, we used a value of 0.1, which is the minimum of the observed water content in the region. We compared PTFs by Rawls and Brakensiek (1985), Wösten et al. (1999), and Weynants et al. (2009) regarding their fit in the model calibration (data following Wösten et al. in Table 1). Saturated hydraulic conductivity $k_s$ (cm s$^{-1}$) had to be estimated separately instead of being taken from the PTFs for two reasons: First, the high variability of hydraulic conductivity for given soil properties makes indirect predictions of $k_s$ through transfer functions error-prone (Dai et al., 2013). Secondly, though macropores have been observed in the field, we did not have the necessary data to explicitly parameterize macropore flow in SWAP. Therefore we estimated lumped effective $k_s$ values in the calibration. Amoozemeter measurements of $k_s$ in the neighborhood of the plot (Table A1) provided the basis for the calibration range of $k_s$.

Plant parameters for grassland were mainly adopted from example input files provided by the model developers. We sampled rooting depths in field measurements and chose the mean as the model parameter. Maximum rooting depths were derived from the soil profiles, assuming that anoxic conditions in the lower horizons restrict root growth (Taylor and Ashcroft, 1972).

We calculated evapotranspiration rates ($ET$) from potential rates following Penman and Monteith (Monteith, 1965), which were split into plant transpiration and soil evaporation by leaf area index (2.5, default). Actual rates were obtained by stress functions for transpiration and by maximum soil water flux and the Boesten–Stroosnijder reduction (Boesten and Stroosnijder, 1986) for soil evaporation, using default parameters (Kroes et al., 2008).

Drainage fluxes through different parts of the model domain are characterized by various resistances which are listed in Tables 1 and 2. Every resistance value can be understood as

$$\text{resistance} = \frac{d}{k_s}$$

where $d$ is the distance which the water passes during drainage (cm). Resistances have, therefore, a unit of time. The values of horizontal drainage and infiltration resistances were chosen based on the distances between model columns and ditches and the averages of observed $k_s$ in the subsoil, i. e. below $-30$ cm (Table 1). The resistance of the aquitard was estimated in the model calibration.

Following Eq. 1, the relevant parameters for salt transport are the dispersion length $L_{\text{dis}}$, the constant groundwater salinity beneath the soil column, the diffusion coefficient, and the root uptake rate. Since diffusion and root uptake are neglectable under the given circumstances (Kroes et al., 2008), they were set to zero a priori.

For the examined site, observations of groundwater level (cm below surface) and electric conductivity $EC$ (mS cm$^{-1}$) over a period of 14 months were available. We calculated salinities from the $EC$ observations by linear regression ($c = 0.54$ mg cm$^{-2}$ mS$^{-1}$ $\cdot EC$) of data from a nearby well (Fig. 1a). In contrast to daily model output, the temporal resolution of the observations was approximately two weeks. In order to reduce the effect of the different resolutions in the calibration and validation, we smoothed the model output by applying a left-sided running two-week mean.



## 2.4 Calibration

We conducted a simple sensitivity analysis for parameters with uncertain or unknown values. The parameters for which the model was most sensitive were chosen for model calibration: saturated hydraulic conductivity $k_s$, dispersion length $L_{dis}$, salinity in the confined aquifer, and vertical aquitard resistance. We estimated these parameters with the PEST software package (Doherty, 2010), using the smoothed model output and the observations of groundwater level and salinity. Comparing calibration results between the different PTFs showed that the functions of Wösten et al. (1999) led to the best reproduction of the medium-term dynamics of both groundwater level and salinity. Multiple calibrations yielded different parameter sets with best accordance of simulated and observed groundwater levels. From those we chose the parameter set for which salinity could also be represented best (Table 2).

## 2.5 Scenarios

In order to study the influence of possible future climate changes, we compared the effects of six different weather scenarios (Table 3) on salinity and water balance components. Data was obtained by the statistical downscaling methods (SDMs) WET-TREG (Spekat et al., 2010) and XDS (Bürger, 1996) in combination with the general circulation models (GCMs) ECHAM5 (Roeckner et al., 2003), ECHAM6 (Stevens et al., 2013), and IPSL-CM5 (Dufresne et al., 2013). Scenarios I and II involve greenhouse gas emission scenarios from the SRES-A2 and SRES-B1 families. In direct comparison, A2 scenarios are characterized by a regionally oriented economical development and a higher global population growth, while B1 scenarios follow a storyline of a more global economic growth and a lower population growth (Nakićenović and Swart, 2000). The other scenarios were based on RCP45 and RCP85 greenhouse gas concentration pathways, corresponding to a radiative forcing of $+4.5$ and $+8.5\,\mathrm{W\,m^{-2}}$, respectively, in the year 2100 compared with pre-industrial values (Moss et al., 2008). SRES emission scenarios were used in the Fourth Assessment Report (AR4) of the Intergovernmental Panel on Climate Change (IPCC), while the RCP scenarios were employed in AR5.

Summer means of the 2011–2100 temperature differ distinctly between SRES scenarios I and II (14.5 and 14.2 °C) and RCP scenarios (between 15.5 and 16.5 °C). Corresponding winter means (all between 5.5 and 6.7 °C) show no similar tendency. Inter-scenario differences between 2011–2100 summer means also occur in annual rainfall, for which I and II have lower values (392.3 and 376.2 mm) than the other scenarios (between 434.1 and 481.9 mm). As well as for temperature, mean annual winter rainfall (between 442.5 and 462.4 mm) is not affected. The 2011–2100 mean of anually summed seasonal rainfall is lower in summer than winter for all scenarios (winter-summer differences between 12.7 and 75.5 mm) except for scenario VI, in which summer rainfall exceeds winter rainfall by 19.5 mm for an average year.

Seasonally averaged temperatures and summed rainfall amounts show trends between 2011 and 2100 in some scenarios (Fig. 2). While seasonal temperatures have significant linear trends between $+0.01$ and $+0.04\,\mathrm{K\,a^{-1}}$ in all scenarios (strongest trends in I, IV, and VI), the trend in rainfall is significant only for the winters of scenarios I ($+0.68\,\mathrm{mm\,a^{-1}}$), IV ($+0.90\,\mathrm{mm\,a^{-1}}$), and VI ($+1.39\,\mathrm{mm\,a^{-1}}$) and for the summers of scenario IV ($-0.60\,\mathrm{mm\,a^{-1}}$) and VI ($+0.92\,\mathrm{mm\,a^{-1}}$). Significance was determined with the Mann-Kendall trend test (significance level 95 %).



Piezometer observations of the deep pressure heads beneath Pewsum near the Freepsumer Meer (Fig. 1a) between 1990 and 2014 showed no significant trend although a trend might be expected due to the observed sea level rise during the same time (Nerem et al., 2018). In addition, including pressure heads simulated with a regional hydrological model would add considerable uncertainty to our scenarios and the results. We therefore selected a one-year window of recorded pressure heads as permanently repeated boundary condition in the calibration.

## 3  RESULTS

### 3.1  Calibration

In the calibrated model results (Fig. 3), both groundwater levels and salt concentrations lie within the observed ranges. The sub-seasonal dynamics are mostly well represented, especially during the winter months. In cases of rain scarcity, the model strongly overestimates salinities and underestimates groundwater levels.

### 3.2  Scenario runs

In all scenario runs, annual means of solute concentration (cf. Fig. 4, left) show constant or increasing salinities below a depth of $-31.5$ cm: Annual mean salinity significantly increased at $-65$ cm in scenarios I, III, IV, and VI (Table 4). At $-31.5$ cm, only scenarios III and IV lead to significant increases. In contrast, annual mean salinity in topsoil layers, i. e. above $-30$ cm, tends to decrease. This trend is significant at $-22$ cm in scenario VI only, while at $-10$ cm scenarios I, IV, and VI show sig-
15 nificant annual-mean decreases. Seasonal means of salinity show a similar behavior as the annual means with mostly stronger declines in summer and stronger increases in winter. Stronger summer declines are even evident in scenario IV, which involves increasing winter and decreasing summer rainfall. These seasonal features in salinity do not show at $-65$ cm, where differences between seasonal-mean trends are dissimilar. In scenario V, only positive winter trends in depths between $-41.5$ and $-48$ cm with comparably low rates are significant. Annual-mean surface salinity decreased in scenarios I, IV, and VI.
Mean salinity at $-65$ cm for the 2000–2099 period is lower in the SRES scenarios I and II than in the RCP scenarios. Conversely at $-10$ cm, it is higher in I and II than in the other scenarios (Table 6). The difference between the SRES/RCP scenarios is also evident in the water balance (Fig. 4, right), where I and II have higher groundwater levels and lower vertical fluxes ($ET$ and bottom flux) than the RCP scenarios.

We quantified the intensity of salt stress on plants as exceedance frequency, i. e. the number of days per year on which daily
mean salinity exceeds a certain value (Table 5). Significant trends in exceedance frequency between 2000 and 2099 resemble the trends in salinity (Table 4). Comparing the 10-year mean frequencies of 2000–2009 and 2090–2099 for each case, we find the strongest trends to occur for 2 and 4 $\mathrm{mg\,cm^{-3}}$ at $-48$ and $-65$ cm, respectively, in most scenarios.

Predictive analyses of the scenario runs using PEST (Doherty, 2010) gave an annual mean prediction uncertainty of $\pm 0.05\,\mathrm{mg\,cm^{-3}}$ at the end of 2099 for all soil layers. All significant trends are at rates outside the uncertainty range and are therefore quite
certain.




## 4   Discussion

Comparing the mean seasonal rainfall in the calibration period and the scenarios (Fig. 2) shows that all scenarios assume wetter summers than the calibration run, and winter rainfall does not decrease in any scenario. Therefore, the overestimation of salinity during dry periods in the calibration (Fig. 3) only results in a small error in the scenario runs.

The results of all scenario runs indicate that rising salinity should not be expected in topsoil layers. Moreover, scenarios with

increasing winter rainfall (I, IV, VI) lead to decreasing topsoil salinity. Although these scenarios include the strongest seasonal temperature increases affecting evapotranspiration, the trend in winter rainfall seems to be a more important, if not the most important factor for the topsoil salt balance. This hypothesis is supported by the fact that salinity declines regardless of the summer rainfall trend (declining in IV, increasing in VI) and the mean summer rainfall (lower than winter in IV, higher in VI).

Of the SRES scenarios (I, II), which both involve an intermediate temperature rise within the range of the RCP scenarios,

only scenario I leads to significant trends in annual mean salinity which are similar yet lower than trends in the other scenarios. Despite involving stronger seasonal temperature increases than scenario III, salinity trends in scenario I are weaker. It is very likely that, through evapotranspiration, the higher mean summer temperature in III amplifies the positive salinity trend in subsoil. Another reason could be that scenario I includes a significant positive trend in winter rainfall, where scenario III does not. Thus, subsoil salinity might be affected by winter rainfall, as was already hypothesized for topsoil salinity. Nevertheless,

scenario I stands out through the high frequency of exceedance of $2\,\mathrm{mg\,cm^{-3}}$ at $-48\,\mathrm{cm}$ (Table 5) despite the similar trend in salinity and a similar mean salinity ($> 2\,\mathrm{mg\,cm^{-3}}$, Table 6) as other scenarios. This suggests a lower sub-annual variability in salt concentration which is therefore another important factor determining salt stress.

Scenarios III and IV strongly differ in the radiative forcing and accordingly cause different salinity trends in the lower soil layers, as well as scenarios V and VI. The main difference between the ECHAM6 and the IPSL-CM5 simulations lies in the

opposed summer rainfall trends of scenarios IV (negative) and VI (positive). Subsoil salinity trends agree with these differences in rainfall trends (cf. IV and VI in Table 4). General differences in the water balance and the vertical salinity gradient between the SRES and RCP scenarios can be attributed to differences in the summer means: Smaller rainfall amounts in the SRES scenarios involve less dilution of upper soil water and thus higher topsoil salinity. Temperature differences might contribute to subsoil salinity, as higher evapotranspiration rates in the RCP scenarios imply a higher bottom flux (Fig. 4, bottom right) and

stronger subsoil salinization.

Plots of water balance (Fig. 4, right) indicate that of the boundary fluxes ($ET$, bottom flux, drainage flux) bottom flux is the most important on the salt balance, since $ET$ is small and drainage flux is governed mainly by bottom flux, rainfall, and groundwater management.

Salinization may impact future land use through changes in plant species communities. In scenario IV, the days per year with

$> 4\,\mathrm{mg\,cm^{-3}}$ increase from an average of $94.7\,\mathrm{d\,a^{-1}}$ in 2000–2009 to $196.1\,\mathrm{d\,a^{-1}}$ in 2090–2099 (Table 5). The occurrence of higher salinities ($> 6\,\mathrm{mg\,cm^{-3}}$) even increases from (almost) 0 (2000–2009) to 9.9 and $31.8\,\mathrm{d\,a^{-1}}$ (2090–2099), in scenarios III and IV, respectively, thus representing a relatively new potential stress on the plant community. However, most scenarios predict salinity increases only in soil layers below $-40\,\mathrm{cm}$, i. e. outside the rooting zone of most grassland species. Therefore,





we do not expect immediate impacts of the salinity changes on the current vegetation, which consists mainly of salt-intolerant grassland species (Table 7).

While we do not expect these effects based on our simulations, future changes of water management would lead to changes of not only groundwater levels but also salinity. The COMTESS project investigates four alternative land management options which include a polder in the Freepsumer Meer where our study plots are located (Karrasch et al., 2017). Should groundwater levels and salinity rise, the species composition on our plots will likely shift toward more salt-tolerant species in the regional species pool. Halophytes overcome osmotic stress with the help of nitrogen-rich osmoprotectants (Steward et al., 1979; Rozema et al., 1985) which lead to their high forage value. Although salt marshes are considered as very productive and providing high-value fodder (Bakker et al., 1993), their productivity and fodder quality depend on salinity level and species composition (Masters et al., 2010; Rogers et al., 2005). In addition, the accumulation of salts might under certain conditions cause depressed feed intake and even compromise animal health (Masters et al., 2007).

Relating to animal health, salinization of ditch water may become a problem for pasture farming if thresholds for drinking water of livestock recommended by the German Federal Ministry for Food and Agriculture (3 mS cm$^{-1}$, Kamphues et al., 2007; 1.62 mg cm$^{-3}$ according to our regression) are exceeded. In this case, farmers need to find logistic solutions for purchasing water, thus reducing the profit from pasture farming. However, our model results of the lateral drainage salinity do not indicate a distinct trend for the 2000–2099 period (not shown). We therefore assume that this case is not likely.

As climate change and sea level rise interact with intensive land use in European coastal landscapes, further research into the influence of increasing salinity in the subsoil on future sustainable land management is necessary.

## 5 Conclusions

Our results suggest that a long-term climate change toward higher temperatures implies slowly rising salt concentrations and thereby rising exceedance frequencies of critical thresholds in the subsoil of the Freepsumer Meer. These trends were mostly significant in projections driven by 4 out of 6 weather scenarios with different trends (winter positive, summer positive; winter positive, summer negative; no trends) of seasonal rainfall. The salinity in the topsoil, however, is expected to decrease or stay constant. We detected the trend in winter rainfall as a main factor affecting the near-surface salt balance. Subsoil salinity trends could be attributed to both winter rainfall and mean summer temperature. Absolute salinity in both top- and subsoil showed a dependence on summer rainfall and temperature through dilution and evapotranspiration, as was shown for substantial differences in climate variables between scenarios. Comparison between one SRES and different RCP scenarios showed that similar mean salinity and salinity trends can result in different magnitudes of salt stress for plants, for which we found positive subsoil trends in most scenarios.

At the soil surface, the projections show only minimal changes in the salt concentrations until 2100. Therefore, land management plans should consider how the strongest changes in salinity, which in our simulations occur between −50 and −60 cm relative to the surface, might affect the vegetation and the use of water. Based on our simulations, we would expect changes in the plant community only if future management should allow groundwater levels to rise. In this case, possible repercussions



on the fodder quality are diverse and difficult to estimate, while pasture farmers might suffer a loss of livestock drinking water from ditches.

*Sample availability.*  Samples of measured saturated hydraulic conductivity ($k_s$), organic carbon content, bulk density, and saturated water content ($\Theta_s$) are available.

## Appendix A:  RECORDED SOIL DATA

*Author contributions.*  J. Eberhard set up the model, conducted simulation runs, did most of the quantitative analysis, and prepared most parts of the text, except for the vegetation-specific part. N.L.M.B. van Schaik helped in the model set up, participated in all qualitative analyses and discussions, and co-authored all text parts. A. Schibalski provided input data, participated in all qualitative analyses and discussions, prepared the texts specific on vegetation, and co-authored all text parts. T. Gräff provided large parts of the input data, participated in all qualitative analyses and discussions, and co-authored all text parts.

*Competing interests.*  No competing interests are present.

*Acknowledgements.*  We thank the COMTESS group (Miguel Cebrian, Juliane Trinogga, Martin Maier, Michael Kleyer) for maintenance, sampling campaigns, and supply; the water board I. Entwässerungsverband Emden, especially Jan van Dyk; and the environmental agency NLWKN field office Aurich (Niedersächsischer Landesbetrieb Wasserwirtschaft, Küsten- und Naturschutz), especially Anke Joritz for data and support. We acknowledge the funding by the German Federal Ministry of Education and Research as part of the collaborative research

project 'Sustainable coastal land management: Trade-offs in ecosystem services COMTESS' (grant number 01LL0911).



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



**Table 1.** Some model parameters.

| Depth of soil layer (cm) | $\Theta_r$ (%) | $\Theta_s$ (%) | $\alpha$ (cm$^{-1}$) | $\lambda$ (–) | $n$ (–) |
|---|---|---|---|---|---|
| *Mualem–van Genuchten parameters ($\alpha$, $\lambda$, $n$ after Wösten et al., 1999)* | | | | | |
| 0 ... 8 | 0.1 | 61.3 | 0.0106 | −0.9150 | 1.4063 |
| 8 ... 24 | 0.1 | 56.5 | 0.0729 | −2.2746 | 1.1133 |
| 24 ... 44 | 0.1 | 58.2 | 0.0556 | −2.0354 | 1.0843 |
| 44 ... 60 | 0.1 | 59.3 | 0.0541 | −1.7990 | 1.0782 |
| 60 ... 100 | 0.1 | 65.0 | 0.2984 | −3.2122 | 1.0836 |

| Mean drain base (cm) | Groundwater level for max. infiltration (cm) | Entry, exit resistance (d) | Drainage, infiltration resistance (d) |
|---|---|---|---|
| *Lateral boundary parameters* | | | |
| −101 | −110 | 1 | 50 |



**Table 2.** Estimated parameters. Calibration ranges are put in brackets.

| Depth of soil layer (cm) | $k_s$ (cm d$^{-1}$) | $L_{dis}$ (cm) |
|---|---|---|
| 0 ... 24 | 20.2 [1.0 ... 100.0] | 25 [1 ... 70] |
| 24 ... 100 | 2.4 [1.0 ... 20.0] | 60 [1 ... 70] |
| Salt concentration in deep aquifer (mg cm$^{-3}$) | 6.6 [6.0 ... 11.0] | |
| Vertical aquitard resistance (d) | 500 [100 ... 500] | |



**Table 3.** General circulation models, statistical downscaling methods, and greenhouse gas scenarios used for computation of the weather scenarios.

| Scen. | GCM | SDM | Greenhouse gas scenario |
|-------|-----|-----|-------------------------|
| I | ECHAM5 | WETTREG | SRES-A2 |
| II | ECHAM5 | WETTREG | SRES-B1 |
| III | ECHAM6 | XDS | RCP45 |
| IV | ECHAM6 | XDS | RCP85 |
| V | IPSL-CM5 | XDS | RCP45 |
| VI | IPSL-CM5 | XDS | RCP85 |





**Table 4.** Significant trends (2000–2099) of annual and seasonal mean salinity in $\text{mg cm}^{-3}\,(100\,\text{a})^{-1}$ in scenario runs, significance level 95 % (*) or 99 % (**). n. s. = not significant.

| Depth (cm) | | I | II | III | IV | V | VI |
|---|---|---|---|---|---|---|---|
| −0.25 | annual | −0.0527** | n. s. | n. s. | −0.0781** | n. s. | −0.0681** |
| | winter | −0.0514** | n. s. | n. s. | −0.0583** | n. s. | −0.0460* |
| | summer | −0.0540* | n. s. | −0.0497* | −0.0978** | n. s. | −0.0902** |
| −10 | annual | −0.0656* | n. s. | n. s. | −0.0986** | n. s. | −0.0896** |
| | winter | −0.0647** | n. s. | n. s. | −0.0791** | n. s. | −0.0642* |
| | summer | −0.0665* | n. s. | n. s. | −0.1180** | n. s. | −0.1149** |
| −22 | annual | n. s. | n. s. | n. s. | n. s. | n. s. | −0.0607** |
| | winter | −0.0470* | n. s. | n. s. | n. s. | n. s. | −0.0547* |
| | summer | n. s. | n. s. | n. s. | n. s. | n. s. | −0.0666** |
| −31.5 | annual | n. s. | n. s. | +0.1117** | +0.1162** | n. s. | n. s. |
| | winter | n. s. | n. s. | +0.1183** | +0.1402** | n. s. | n. s. |
| | summer | n. s. | n. s. | +0.1052* | +0.0923* | n. s. | n. s. |
| −41.5 | annual | +0.0863* | n. s. | +0.1559** | +0.1814** | n. s. | n. s. |
| | winter | +0.0996** | n. s. | +0.1645** | +0.2119** | +0.0889** | n. s. |
| | summer | n. s. | n. s. | +0.1474* | +0.1509** | n. s. | n. s. |
| −48 | annual | +0.1356** | n. s. | +0.1959** | +0.2399** | n. s. | n. s. |
| | winter | +0.1559** | n. s. | +0.2173** | +0.2930** | +0.1205** | n. s. |
| | summer | n. s. | n. s. | n. s. | +0.1870** | n. s. | n. s. |
| −65 | annual | +0.4291** | n. s. | +0.5734** | +0.8729** | n. s. | +0.4504** |
| | winter | +0.4097** | +0.1863* | +0.5345** | +0.8969** | n. s. | +0.3934* |
| | summer | +0.4483** | n. s. | +0.6120** | +0.8491** | n. s. | +0.5070** |
| −95 | annual | n. s. | n. s. | +0.0572** | +0.1027** | n. s. | n. s. |
| | winter | n. s. | n. s. | +0.0777** | +0.1474** | n. s. | n. s. |
| | summer | n. s. | n. s. | n. s. | +0.0583** | n. s. | n. s. |





**Table 5.** Significant trends in exceedance frequency (days per year), significance level 95 % (*) or 99 % (**). Values in parentheses indicate mean frequencies for the periods of 2000–2009 and 2090–2099, respectively. n. s. = not significant.

| Salinity (mg cm$^{-3}$) | Depth (cm) | I | II | III |
|---|---|---|---|---|
| 2 | −41.5 | n. s. | n. s. | (19.9; 45.2)* |
|   | −48 | (266; 337.3)** | n. s. | (192.2; 289.7)** |
| 4 | −65 | (31.4; 125.6)** | n. s. | (98.1; 163.3)** |
| 6 | −65 | n. s. | n. s. | (0; 9.9)** |

| Salinity (mg cm$^{-3}$) | Depth (cm) | IV | V | VI |
|---|---|---|---|---|
| 2 | −41.5 | (9.7; 51.6)** | n. s. | n. s. |
|   | −48 | (182.6; 345.9)** | (200.5; 296.6)* | n. s. |
| 4 | −65 | (94.7; 196.1)** | n. s. | (128.2; 174.6)** |
| 6 | −65 | (2.7; 31.8)** | n. s. | (11.7; 20.3)* |



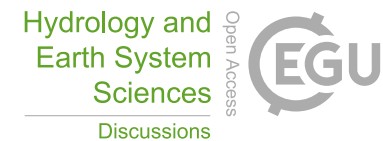

**Table 6.** 2000–2099 mean salinity ($\mathrm{mg\,cm^{-3}}$) in scenario runs.

| Depth (cm) | I | II | III | IV | V | VI |
|---|---|---|---|---|---|---|
| −10 | 0.5331 | 0.5573 | 0.4215 | 0.4205 | 0.4389 | 0.4195 |
| −31.5 | 1.4049 | 1.4318 | 1.3753 | 1.3961 | 1.3166 | 1.3738 |
| −48 | 2.1963 | 2.2218 | 2.1858 | 2.2288 | 2.0802 | 2.2418 |
| −65 | 3.7603 | 3.7213 | 4.1521 | 4.1861 | 3.9296 | 4.1859 |





**Table 7.** Salt indicator values occurring on the two sample plots (Cebrián-Piqueras et al., 2017).

| Species | Ellenberg salt indicator value (S)[1] |
|---|---|
| *Agrostis stolonifera* L. | 0 – glycophyte |
| *Alopecurus geniculatus* L. | 2 – oligohaline ($0.05$–$0.3\,\%\,Cl^-$) |
| *Alopecurus pratensis* L. | 0 – glycophyte |
| *Dactylis glomerata* L. | 0 – glycophyte |
| *Elymus repens* (L.) Gould | 0 – glycophyte |
| *Festuca rubra* L. | 0 – glycophyte |
| *Glyceria fluitans* (L.) R.Br. | 0 – glycophyte |
| *Lolium perenne* L. | 0 – glycophyte |
| *Phragmites australis* (Cav.) Trin. ex Steud. | 0 – glycophyte |
| *Ranunculus repens* L. | 1 – salt-tolerant ($\leq 1.9\,mg\,cm^{-3}$) |
| *Trifolium repens* L. | 1 – salt-tolerant ($\leq 1.9\,mg\,cm^{-3}$) |

[1] Ellenberg and Leuschner (2010)





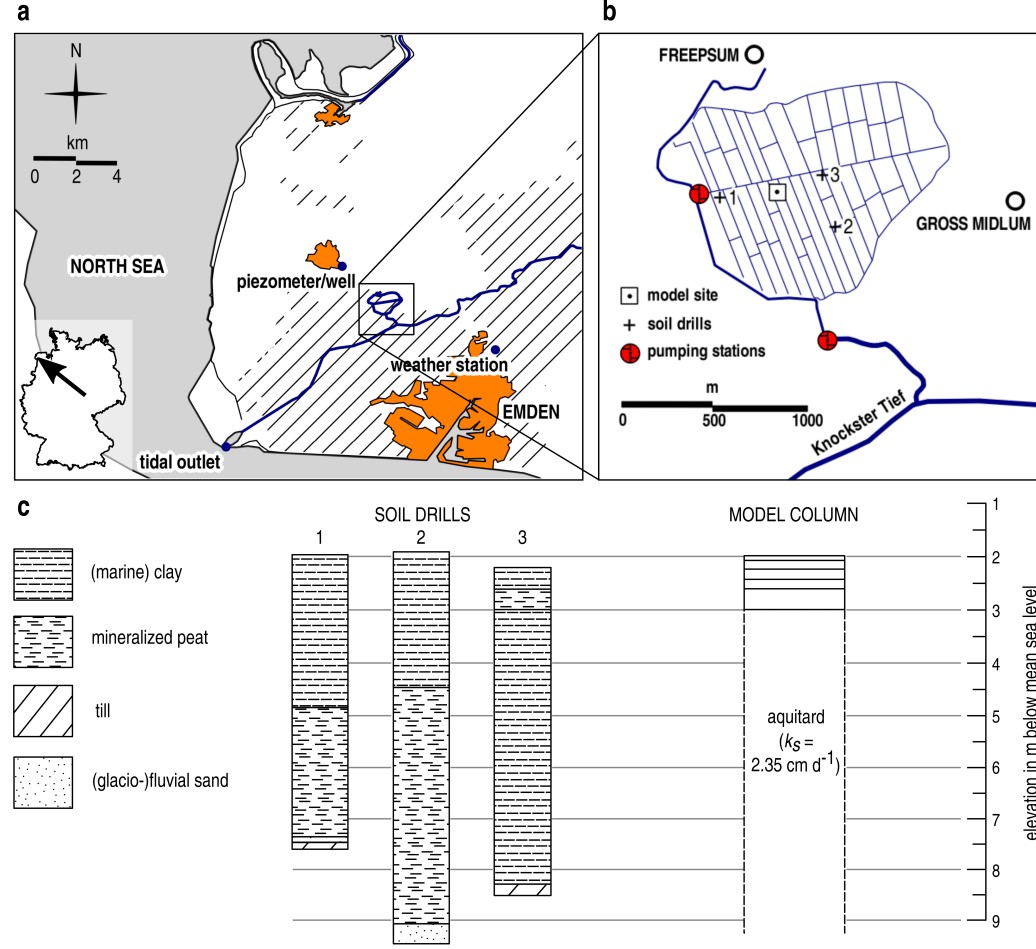

**Figure 1.** Overview of the study area. (a): Location of the Freepsumer Meer within the Krummhörn region; blue lines: main water courses, including the Knockster Tief, connecting it to the North Sea through the tidal outlet; thin black lines along the coast: dikelines; orange areas: major settlements; hatched areas: surface beneath mean sea level. (b): Location of the model site, soil drills, and drainage structure of the Freepsumer Meer; surface drains are elevated according to their thickness: lowest drains the thinnest lines, highest the thickest. (c): Soil drills representing the geological structure of the region: clay layers on top of (glacio-)fluvial sand with intermediate peat (Wildvang, 1938); on the right, the model column as employed in the model.





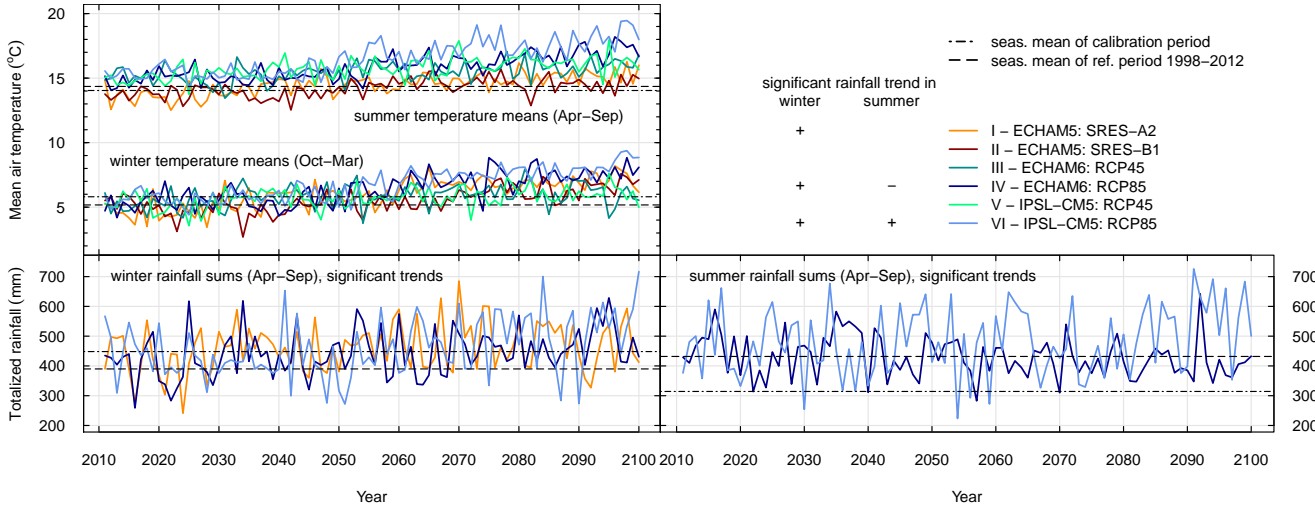

**Figure 2.** Seasonal mean temperatures and rainfall sums in the scenarios I to VI (2011–2100). Comparison between the means of the calibration period (10/2011–12/2012), the means of a reference period (1998–2012), and the scenario inputs. Only time series with significant trends (Mann-Kendall test, $p < 5\,\%$) are shown. All temperature scenarios have significant positive trends. Winter means were calculated for the periods from October to March, summer means from April to September.





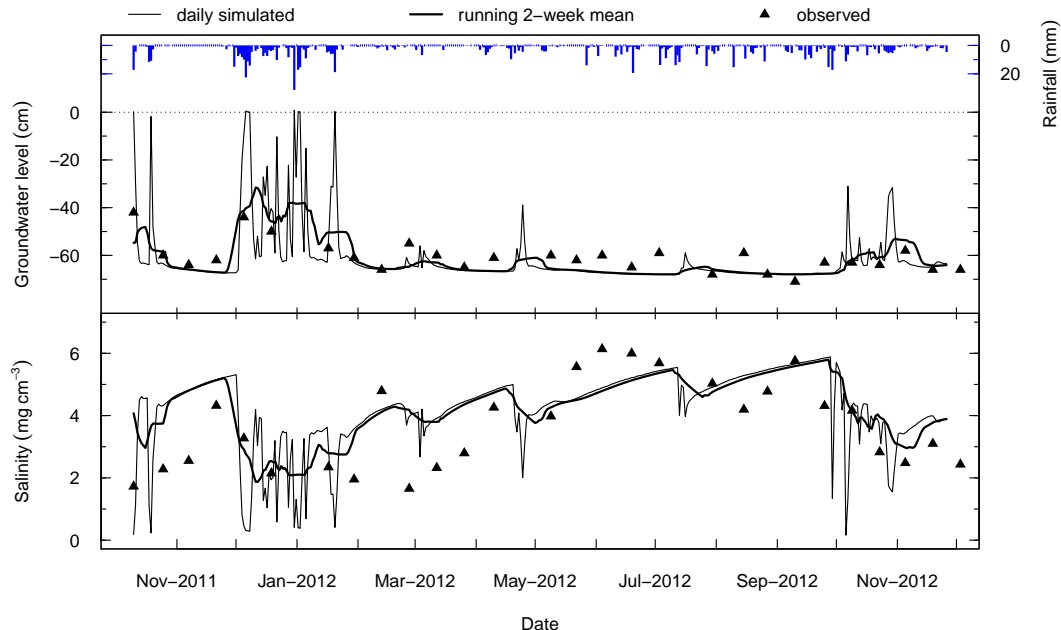

**Figure 3.** Results of the calibrated model for groundwater level and salinity at groundwater level.

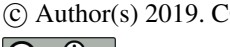



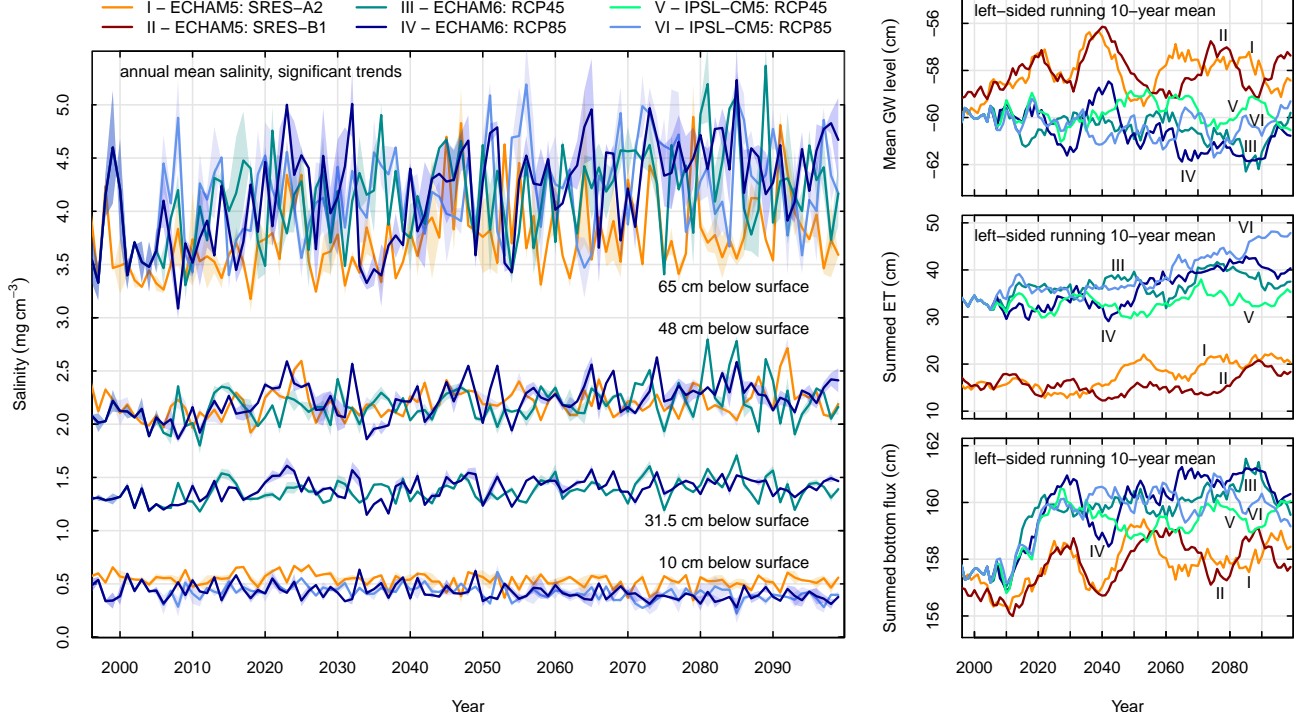

**Figure 4.** A selection from the annual output of the scenario runs. Left: Annual mean salinity shows positive or no (not shown) trends in layers at $-31.5$ cm and below, whereas the upper layers above $-22$ cm have negative or no (not shown) trends. Buffers around the graphs indicate the annual ranges between winter (lower) and summer (higher salinities) means. Right: Annual mean (groundwater level) and annually summed ($ET$, bottom flux) water balance components indicate a strong dependence of the groundwater level on the bottom flux since $ET$ fluxes are comparably small. Note that water balance components are given in cm.


**Table A1.** Amoozemeter measurements (sample size = 3) of $k_s$ (cm d$^{-1}$).

| Mean depth (cm) | minimum | maximum | mean | standard deviation |
|---|---|---|---|---|
| 22 | 20.22 | 93.31 | 54.00 | 36.89 |
| 51 | 6.47 | 16.50 | 10.11 | 5.58 |
| 70 | 1.21 | 12.79 | 6.58 | 5.83 |





**Table A2.** Field data of soil physical properties. $N$ = sample size, s. d. = sample standard deviation, n. d. = not defined.

| Depth (cm) | N | Organic carbon (%) | | | | Bulk density (g cm$^{-3}$) | | | | Sat. water content $\Theta_s$ (%) | | | |
|---|---|---|---|---|---|---|---|---|---|---|---|---|---|
| | | min | max | mean | s. d. | min | max | mean | s. d. | min | max | mean | s. d. |
| 0 … 8 | 1 | 1.50 | 1.50 | 1.50 | n. d. | 0.78 | 0.78 | 0.78 | n. d. | 70.7 | 70.7 | 70.7 | n. d. |
| 8 … 24 | 7 | 0.25 | 14.06 | 10.21 | 4.73 | 0.68 | 1.30 | 0.83 | 0.22 | 51.1 | 72.0 | 66.4 | 7.0 |
| 24 … 44 | 7 | 0.00 | 10.21 | 6.36 | 3.46 | 0.78 | 1.12 | 0.88 | 0.12 | 57.9 | 68.0 | 64.8 | 3.6 |
| 44 … 60 | 7 | 0.00 | 10.21 | 6.36 | 3.46 | 0.78 | 0.93 | 0.85 | 0.06 | 62.0 | 68.0 | 65.9 | 1.9 |
| 60 … 100 | 13 | 0.00 | 32.23 | 12.39 | 9.68 | 0.21 | 0.76 | 0.57 | 0.21 | 69.0 | 90.0 | 76.4 | 7.5 |