# Peer review of "Simulating future salinity dynamics in a coastal marshland under different climate scenarios"

_Hydrology and Earth System Sciences, 2018_

## Referee Comment (RC1) · Anonymous Referee #1 · 19 Feb 2019

The authors Eberhard et al. used a one-dimensional SWAP model to predict the salinity variation in the next 100 years at a coastal marshland. Six climate conditions are considered in this prediction, the effect of rainfall and temperature on the salinity variation are investigated. This model still needs to be improved, model setup should be clarified and some figures should be improved. Currently, this study is too thin to publish on HESS. In my point of view, more simulations or models need to be added to this study until it is ready to be published.

1. The content or scientific significant is not enough. As the authors mentioned in Section 4, many factors may change in the future, only considering salt from deeper aquifer seems not well considered. I suggest adding more simulations based on this model. The effect of short term scenarios is a good idea, such as the storm surges

which authors mentioned in Section 1 or extreme long dry season. Also, sea level rise as a global concern should be considered.

2. Please stress the novelties of this study at the end of Section 1.

3. Page 4, Line 20-24, authors illustrate the boundary conditions. Please clarify the boundary conditions for salinity, how much salt comes from the bottom boundary?

4. Page 4, Line 33-34, authors used grain size distribution, organic carbon content, and bulk density to estimate three parameters. Please clarify how this estimation works, any equations used in this estimation. Authors can provide some supplementary of this estimation if necessary.

5. Page 6 and Page 7, authors illustrate the calibration and provide Fig 3 to prove a good fit between simulation results and observations. Please provide more details about this calibration. The calibration results should be quantified to make this calibration more persuasive to readers, using correlation coefficient or Nash–Sutcliffe coefficient or other methods to quantify the comparison between simulation and observation. In addition, I suggest briefly introduce the basic theory of PEST to make readers understand this process more clearly.

6. Page 6, Line 10-20, please provide more details about these six scenarios, such as what's the difference between SRES and RCP. These climate conditions are cited from other literature, but still need to illustrate in this study to make readers understand the predictions clearly.

7. Page 5, line 21, is this equation 2? Please mark clearly.

8. Table 1 is too simple to understand, the figure and table should be complete and informative itself. Please add more illustration in the title, and clarify what is $\Theta r$, $\Theta s$, and other symbols. Please amend it for other tables and figures.

9. Please improve or redraw Fig 1 a and b, the study site seems not clear. In Fig 1a, please use some color instead of the hatched area; mark that the small map on the left

corner is German. In Fig 1b, please adding elevation data in the study site. I am not sure what's the blue lines in the lake in Fig 1b, do these blue lines represent drains? Why there are so many drains in this former lake?

10. Figure 2, the plot of Totalized Rainfall to year, the winter rainfall sums should be Oct-Mar.

---

## Referee Comment (RC2) · Anonymous Referee #2 · 27 Feb 2019

Eberhard et al. evaluated the salinity dynamics in the coastal marshland using the one-dimensional SWAP model, and then simulate its future impacts with a group of GCMs under climate change scenarios. The authors also quantitatively estimated the salinity in each scenario and evaluated the seasonal sensitivities of salinity trends. In general, the science is novel, and the paper is well written. However, I'm not quite convinced by the SWAP model application in this specific study, and more details are required to demonstrate the model accuracy. I recommend a major revision, or even reject considering the time needed for improving this paper. Here are some of my comments.

A detailed introduction of marshland salinization is missing, as the authors only generally discussed the seawater intrusion and salinization. Why marshland salinization is

important to study? What are the science questions and difficulties in modeling marsh-land salinization? More parameter information is required, for example, how is the heterogeneity of marshland being solved in the 1D SWAP model? How is the ground-water level simulated in the model, as SWAP only solves the flow in the soil? Did you simulate the horizontal water flux? In general, I'm not convinced by the modeling ca-pability of 1D SWAP model to accurately simulate salinization in the marshland. It's not clear how salinization is simulated in the model. Usually, seawater intrusion in the aquifer is simulated by a coupled variable-density flow and solute transport processes, or equivalent freshwater head calculated by the salinity and the depth of aquifer. Did you consider this? Anyway, I didn't find how salinization is simulated in this paper. The subsection 2.4 calibration and 3.1 calibrations are confusing. The results are not suffi-ciently discussed. For example, what's the initial condition did you use? And, it seems salinity is measured at soil layer/unconfined layer, and simply diluted by the rain water? Is salinity in the confined layer a boundary condition? How is it determined? How do bottom flux and drainage flue being calculated? The parameters being perturbed in the scenario runs are not clear. I assume temperature and rainfall are obvious, but did you change the radiation? Does the potential sea level rise being considered? If sea level does not rise, how does wetter climate increase salinity? More explanations are required.

---

## Author Comment (AC1) · 14 Mar 2019

We thank Anonymous Referee #1 for reviewing our manuscript. We believe that the comments and suggestions identified important issues and clearly help to improve the paper, which we are very grateful about. In the following, we would like to respond point by point to the referee comments (RC), typeset in italic type, to the best of our abilities. Responses are marked as author comments (AC) and typeset in roman type.

**RC1**: *The content or scientific significant is not enough. As the authors mentioned in Section 4, many factors may change in the future, only considering salt from deeper aquifer seems not well considered. I suggest adding more simulations based on this*

*model. The effect of short term scenarios is a good idea, such as the storm surges which authors mentioned in Section 1 or extreme long dry season. Also, sea level rise as a global concern should be considered.*

**AC1**: Thank you for the suggestions. We agree that the future development of coastal marshland salinization can depend on multiple factors. Considering the geological setting, in this study area the slow upward seepage from a deep aquifer through a thick confining layer is expected to be the main mechanism of salinization. Direct lateral intrusion of salt water to the unconfined aquifer is not an issue here. Regarding storm surges, there are no known floodings in the past which might have affected the salinity in the Freepsumer Meer; the area is furthermore surrounded by dikes, making flooding due to a storm surge rather unlikely in the future. Through the use of different climate change scenarios, we believe to have included different long-term climatic conditions within the expected ranges and have analyzed their effects. The issue of sea level rise has been approached on page 7, lines 1–5. It has not been considered in the study due to two reasons. First, observations of the deep pressure head nearby the study site did not exhibit any significant trends in 1990–2014 despite an observed sea level rise in this period. Second, we regarded simulated deep pressure heads near the study site, produced by the hydrological model GSFLOW, driven by the climate scenarios I and II. The pressure head differences between simulations with assumed 0 cm, 80 cm, and 150 cm sea level rise within the 2000–2100 period were marginal (in the order of 10 cm at the end of the period) and the uncertainties involved in the model setup were considerable. We concluded that the effect of sea level rise is negligible for salinization in our case. We will include more explanations on the expected relevance of these different processes for the salinization in our study area and the subsequent choice of the model in our introduction when we revise the paper.

**RC2**: *Please stress the novelties of this study at the end of Section 1.*
**AC2**: We acknowledge that the study's novelties are not sufficiently communicated in the introduction. In a revised version of the manuscript, we would therefore add

the following point: The study is novel in that we concentrate on long-term climatic effects on slow salinization from upward seepage of deep groundwater, which we expect to be the main mechanism in the study site. This contrasts with a number of studies describing direct sea-water intrusion or short-term effects through boils, paleochannels, etc. in coastal marshlands (e.g. Weerts, 1996; Kim et al., 2003; de Louw et al., 2010; Colombani et al., 2015; Kliesch et al., 2016).

**RC3**: *Page 4, Line 20-24, authors illustrate the boundary conditions. Please clarify the boundary conditions for salinity, how much salt comes from the bottom boundary?*
**AC3**: Indeed, a specification of the bottom boundary condition for salinity is missing and will be added. The salinity of the confined groundwater is given as a constant parameter in the model and was estimated in the model calibration. Table 2 in the manuscript provides the calibration range and the estimated value of $6.6\,\mathrm{mg\,cm^{-3}}$.

**RC4**: *Page 4, Line 33-34, authors used grain size distribution, organic carbon content, and bulk density to estimate three parameters. Please clarify how this estimation works, any equations used in this estimation. Authors can provide some supplementary of this estimation if necessary.*
**AC4**: Thank you for the suggestion. We agree that the manuscript can be improved by adding the proposed details. As discussed in Section 2.4, we used pedotransfer functions of Wösten et al. (1999) in the calibrated model (with an adjustment of $\Theta_r$), viz.

$\lambda = 10\frac{\exp(\lambda^*)-1}{\exp(\lambda^*)+1},$

$\alpha = \exp\{-14.96 + 0.03135 \times \mathsf{clay} + 0.0351 \times \mathsf{silt} + 0.646 \times \mathsf{SOM} + 15.29 \times \mathsf{BD}$
$\qquad - 0.192 \times \mathsf{topsoil} - 4.671 \times \mathsf{BD}^2 - 0.000781 \times \mathsf{clay}^2 - 0.00687 \times \mathsf{SOM}^2$
$\qquad + 0.0449/\mathsf{SOM} + 0.0663 \times \ln(\mathsf{silt}) + 0.1482 \times \ln(\mathsf{SOM}) - 0.04546 \times \mathsf{BD} \times \mathsf{silt}$
$\qquad - 0.4852 \times \mathsf{BD} \times \mathsf{SOM} + 0.00673 \times \mathsf{topsoil} \times \mathsf{clay}\}\,\mathrm{cm^{-1}},$

$n = 1 + \exp\{-25.23 - 0.02195 \times \mathsf{clay} + 0.0074 \times \mathsf{silt} - 0.1940 \times \mathsf{SOM} + 45.5 \times \mathsf{BD}$

$$- 7.24 \times \text{BD}^2 + 0.0003658 \times \text{clay}^2 + 0.002885 \times \text{SOM}^2 - 12.81/\text{BD}$$
$$- 0.1524/\text{silt} - 0.01958/\text{SOM} - 0.2876 \times \ln(\text{silt}) - 0.0709 \times \ln(\text{SOM})$$
$$- 44.6 \times \ln(\text{BD}) - 0.02264 \times \text{BD} \times \text{clay} + 0.0896 \times \text{BD} \times \text{SOM}$$
$$+ 0.00718 \times \text{topsoil} \times \text{clay}\},$$

$\Theta_r = 0.1$,

where clay, silt, SOM denote percentages of clay, silt and soil organic matter, BD is the numerical value (without unit) of bulk density measured in $\text{g cm}^{-3}$, topsoil is 0 (below $-30$ cm) or 1 (above or at $-30$ cm), and

$$\lambda^* = 0.0202 + 0.0006193 \times \text{clay}^2 - 0.001136 \times \text{SOM}^2 - 0.2316 \times \ln(\text{SOM})$$
$$- 0.03544 \times \text{BD} \times \text{clay} + 0.00283 \times \text{BD} \times \text{silt} + 0.0488 \times \text{BD} \times \text{SOM}.$$

We propose to add the equations in a supplementary.

**RC5**: *Page 6 and Page 7, authors illustrate the calibration and provide Fig 3 to prove a good fit between simulation results and observations. Please provide more details about this calibration. The calibration results should be quantified to make this calibration more persuasive to readers, using correlation coefficient or Nash–Sutcliffe coefficient or other methods to quantify the comparison between simulation and observation. In addition, I suggest briefly introduce the basic theory of PEST to make readers understand this process more clearly.*

**AC5**: Thank you for the valuable suggestion. We agree that the calibration needs some clarification. We propose to add the following description to our paragraph on the model calibration:

"The parameter calibration was performed in three steps: (1) First, the parameters saturated hydraulic conductivity ($k_s$), dispersion length ($L_{dis}$), deep groundwater salinity, and vertical resistance were estimated with the aim to minimize the sum of squared deviations between the simulated and measured groundwater levels. For this purpose we used the PEST software package (Doherty, 2010) which uses a steepest decent searching optimization algorithm based on Gauss–Marquardt–Levenberg.

(2) Second, the value of the deep groundwater salinity was varied, leaving all other parameters as estimated in the previous step. Here a visual comparison of measured and simulated salinity was used for the optimization. (3) After the previous steps were performed for each of the three PTFs, the sub-annual dynamics of each were compared (locations of local minima and maxima, ranges of sub-monthly fluctuations) and the PTF of Wösten et al. (1999) was chosen. For the final calibrated model, the sum of squared deviations between modeled and observed groundwater levels is $694.8\,\mathrm{cm}^2$. The mean deviation per observation is $2.0\,\mathrm{cm}$. The correlation coefficient of modeled and observed groundwater levels is 0.76."

**RC6**: *Page 6, Line 10-20, please provide more details about these six scenarios, such as what's the difference between SRES and RCP. These climate conditions are cited from other literature, but still need to illustrate in this study to make readers understand the predictions clearly.*
**AC6**: We will try to make the descriptions on the six scenarios clearer by rephrasing the paragraph about the climate scenarios:

"Scenarios I and II involve greenhouse gas emission scenarios from the SRES-A2 and SRES-B1 families. In direct comparison, A2 scenarios are characterized by a regionally oriented economical development and a higher global population growth, while B1 scenarios follow a storyline of a more global economic growth and a lower population growth (Nakićenović and Swart, 2000). In contrast to the SRES scenarios, in the RCP scenarios the greenhouse gas emissions are decoupled from socioeconomic models. Instead a range of possible future $CO_2$ emissions and radiative forcing trends till 2100 are used. Thus the RCP4.5 and RCP8.5 scenarios correspond to an increase in radiative forcing of 4.5 and $8.5\,\mathrm{W\,m}^{-2}$, respectively, in 2100 compared to pre-industrial values. The differences in the trends in temperature and precipitation for the resulting six scenarios are described in the following paragraphs."

**RC7**: *Page 5, line 21, is this equation 2? Please mark clearly.*

**AC7**: We agree that the typesetting of the second equation may be irritating. Since this equation is not further discussed or cited and was just meant to clarify the meaning of the different resistance values, we suggest including it in the line, e.g.: "Every resistance value can be understood as the ratio $d/k_s$, where $d$ is the distance ..."

**RC8**: *Table 1 is too simple to understand, the figure and table should be complete and informative itself. Please add more illustration in the title, and clarify what is $\Theta_r$, $\Theta_s$, and other symbols. Please amend it for other tables and figures.*

**AC8**: Thank you for this remark. We realize that we should improve the titles of various tables and figures in the manuscript, we will work on this.

**RC9**: *Please improve or redraw Fig 1 a and b, the study site seems not clear. In Fig 1a, please use some color instead of the hatched area; mark that the small map on the left corner is German[y]. In Fig 1b, please adding elevation data in the study site. I am not sure what's the blue lines in the lake in Fig 1b, do these blue lines represent drains? Why there are so many drains in this former lake?*

**AC9**: We see that Figures 1a and b need improvements in order to characterize the study site better. The map of Germany will be designated clearly and the hatched areas will receive a new filling or color in future versions of the manuscript. In fact, the Freepsumer Meer, which is the drained area between the two villages, forms a quite distinctive depression compared to the surrounding areas. We intend to add three indications to the figure caption: First, a more specific description of the extent of the former lake; second, that the whole of the Freepsumer Meer lies on average 1.5 m below the surrounding area; third, a clarification that all blue lines are drains. In our opinion, the second point would avoid the need for detailed elevation data, which might clutter the figure. The relative depression of the Freepsumer Meer also explains why there are so many drains – historically, the drainage of the lake required a fast

and efficient removal of water.

**RC10**: *Figure 2, the plot of Totalized Rainfall to year, the winter rainfall sums should be Oct–Mar.*
**AC10**: Thank you for this correction. It should indeed be "Oct–Mar", we will change it.

**References**

Weerts, J.: Complex confining layers. Architecture and hydraulic properties of Holocene and Late Weichselian deposits in the fluvial Rhine-Meuse delta, the Netherlands, Ph.D. thesis, University of Utrecht, 1996.

Wösten, J., Lilly, A., Nemes, A., and le Bas, C.: Development and use of a database of hydraulic properties of European soils, Geoderma, 90, 169–185, 1999.

Nakićenović, N. and Swart, R.: Special Report on Emissions Scenarios: A special report of Working Group III of the Intergovernmental Panel on Climate Change, Cambridge University Press, 2000.

Kim, Y., Lee, K.-S., Koh, D.-C., Lee, D.-H., Lee, S.-G., Park, W.-B., Koh, G.-W., and Woo, N.-C.: Hydrogeochemical and isotopic evidence of groundwater salinization in a coastal aquifer: a case study in Jeju volcanic island, Korea, Journal of Hydrology, 270, 282–294, https://doi.org/10.1016/S0022-1694(02)00307-4, 2003.

de Louw, P., Oude Essink, G., Stuyfzand, P., and van der Zee, S.: Upward ground-water flow in boils as the dominant mechanism of salinization in deep polders, The Netherlands, Journal of Hydrology, 394, 494–506, 2010.

Doherty, J.: PEST: Model independent parameter estimation, User manual, Water-mark Numerical Computing, Brisbane, 2010.

Colombani, N., Mastrocicco, M., and Giambastiani, B.: Predicting salinization trends in a lowland coastal aquifer: Comacchio (Italy), Water Resource Management, 29, 603–618, 2015.

Kliesch, S., Behr, L., Salzmann, T., and Miegel, K.: Simulation des Grundwasser-haushalts in ausgewählten Niederungsgebieten an der deutschen Ostseeküste, Hy-drologie und Wasserbewirtschaftung, 60, 108–118, 2016.

---

## Author Comment (AC2) · 15 Mar 2019

We thank Anonymous Referee #2 for reviewing our manuscript. We believe that the comments and suggestions identified important issues and clearly help to improve the paper, which we are very grateful about. In the following, we would like to respond point by point to the referee comments (RC), typeset in italic type, to the best of our abilities. Responses are marked as author comments (AC) and typeset in roman type.

**RC1**: *A detailed introduction of marshland salinization is missing, as the authors only generally discussed the seawater intrusion and salinization. Why marshland salinization is important to study? What are the science questions and difficulties in*

*modeling marshland salinization?*

**AC1**: We see now that the second paragraph in the introduction is not very well structured. Though the different possible processes leading to salinization are all mentioned, we see that it can be confusing and not clear which processes are most important in our study area and why we decided to model these processes with SWAP. We will attempt to rewrite these parts of the manuscript in order to clarify the following points: "Marshlands are important for agricultural use and are prone to salinization effects of salt water intrusion from deeper aquifers, especially under the conditions of climate change (de Louw et al., 2010; 2011; 2013; Oude Essink et al., 2010; Herbert et al., 2015). That is because of the inhomogeneous geological setup which consists of Holocene peats, clays, and sand structures in varying thicknesses. Apart from salt import from deeper layers, the salt concentration in a soil increases through evaporation and plant transpiration, whereas it decreases through precipitation and subsequent infiltration (de Louw et al., 2013). So it is important to understand how these types of soils react to changing meteorological conditions with a potential increase in dryness and higher temperatures."

**RC2**: *More parameter information is required, for example, how is the heterogeneity of marshland being solved in the 1D SWAP model?*

**AC2**: Thank you for the suggestion. The horizontal marshland heterogeneity is not resolved in our model. In this model we calibrated the salt transport for a possible profile in the Freepsumer Meer. Of course there are some spatial heterogeneities in the soil profiles in the area, such as minor differences in the surface altitude or the depth of the confining layers. These heterogeneities might also have some impact on the current salinity of the soil water. In this study, however, the aim was to simulate future changes due to climate change. We do not expect that the long-term trend in salinity due to climate change would differ significantly across the area. Therefore we think that the modeled profile can be seen as representative.

**RC3**: *How is the groundwater level simulated in the model, as SWAP only solves the flow in the soil?*

**AC3**: We agree that this point needs clarification in the manuscript. The simulated groundwater level is, after its initialization, determined by the given boundary conditions. That is, we specified pressure heads in the deep aquifer, implying a Dirichlet-type bottom boundary condition for the modeled soil water column. The lateral boundary condition was given by prescribed water levels in the nearby drain ditches and the geometry and hydraulic properties (lateral boundary parameters in Table 1 of the manuscript) of the drained field in which the modeled site is located. Additional fluxes from or toward the top of the soil column are subject to the meteorological conditions. We will add the missing information in the manuscript.

**RC4**: *Did you simulate the horizontal water flux?*

**AC4**: The horizontal water flux was not explicitly simulated. The horizontal fluxes implicitly included in the model are the surface runoff and the drainage flux from or toward the soil column. The latter results from the lateral boundary conditions as specified in AC3.

**RC5**: *In general, I'm not convinced by the modeling capability of 1D SWAP model to accurately simulate salinization in the marshland. It's not clear how salinization is simulated in the model. Usually, seawater intrusion in the aquifer is simulated by a coupled variable-density flow and solute transport processes, or equivalent freshwater head calculated by the salinity and the depth of aquifer. Did you consider this? Anyway, I didn't find how salinization is simulated in this paper.*

**AC5**: Thank you for the suggestions. We would like to clarify the hydrological situation of the model site and why in our view the model is a suitable choice. Our study site, the Freepsumer Meer, consists of numerous rather small fields (cf. Figure 1b of the manuscript), which are separated by drain ditches. Thus, any horizontal groundwater

fluxes are largely dominated by the surface drainage. Direct lateral intrusion of salt water to the unconfined aquifer is not an issue here. As a result, the main mechanism causing the observed salinization must be the upward seepage through the aquitard, the existence of which we infer from the geological setting (Figure 1c). Therefore, the water and salt balance are determined by the one-dimensional bottom flux through the aquitard, the lateral drainage, and the climate conditions at the top. All of these components are considered in our model. Moreover, the SWAP model is designed specifically for the simulation of such field-scale processes and was originally applied in similar landscapes (e.g. Kroes et al., 2000). We will make this point clearer in the manuscript.

**RC6**: *The subsection 2.4 calibration and 3.1 calibrations are confusing.*
**AC6**: We agree that the calibration needs some clarification. We propose to add the following description to subsection 2.4: "The parameter calibration was performed in three steps: (1) First, the parameters saturated hydraulic conductivity ($k_s$), dispersion length ($L_{dis}$), deep groundwater salinity, and vertical resistance were estimated with the aim to minimize the sum of squared deviations between the simulated and measured groundwater levels. For this purpose we used the PEST software package (Doherty, 2010) which uses a steepest decent searching optimization algorithm based on Gauss–Marquardt–Levenberg. (2) Second, the value of the deep groundwater salinity was varied, leaving all other parameters as estimated in the previous step. Here a visual comparison of measured and simulated salinity was used for the optimization. (3) After the previous steps were performed for each of the three PTFs, the sub-annual dynamics of each were compared (locations of local minima and maxima, ranges of sub-monthly fluctuations) and the PTF of Wösten et al. (1999) was chosen."

Additionally, subsection 3.1 will be expanded by a quantification of the calibrated model output: The sum of squared deviations between modeled and observed groundwater levels is $694.8 \, \mathrm{cm}^2$. The mean deviation per observation is 2.0 cm. The correlation

coefficient of modeled and observed groundwater levels is 0.76.

**RC7**: *The results are not sufficiently discussed. For example, what's the initial condition did you use? And, it seems salinity is measured at soil layer/unconfined layer, and simply diluted by the rain water? Is salinity in the confined layer a boundary condition? How is it determined? How do bottom flux and drainage flue being calculated?*

**AC7**: We appreciate the suggestions and agree that some information in the discussion is missing. The simulation period of 2000–2099, which is considered in the results and discussion sections, was preceded by the spin-up period 1961–1999. At 1 January 1961, groundwater level was initialized with the first value of the observed groundwater levels in 2011 ($-42$ cm). The salt concentration profile was initialized with a constant value of $3\,\mathrm{mg\,cm}^{-3}$. Both the groundwater level and the salinity undergo rapid changes and are considered independent from the chosen initial conditions after a few years. The salinity of the confined groundwater is given as a constant boundary condition in the model and was estimated in the model calibration. Table 2 in the manuscript provides the calibration range and the estimated value of $6.6\,\mathrm{mg\,cm}^{-3}$. Bottom and lateral fluxes are calculated as provided in AC3.

**RC8**: *The parameters being perturbed in the scenario runs are not clear. I assume temperature and rainfall are obvious, but did you change the radiation?*

**AC8**: Thank you for the important question. We agree that the parameters modified in the climate scenarios were not sufficiently specified. The scenarios cover time series of radiation, minimum and maximum temperature, air humidity, wind speed, and rainfall.

**RC9**: *Does the potential sea level rise being considered? If sea level does not rise, how does wetter climate increase salinity?*

**AC9**: The issue of sea level rise has been approached on page 7, lines 1–5. It has
not been considered in the study due to two reasons. First, observations of the deep pressure head nearby the study site did not exhibit any significant trends in 1990–2014 despite an observed sea level rise in this period. Second, we regarded simulated deep pressure heads near the study site, produced by the hydrological model GSFLOW, driven by the climate scenarios I and II. The pressure head differences between simulations with assumed 0 cm, 80 cm, and 150 cm sea level rise within the 2000–2100 period were marginal (in the order of 10 cm at the end of the period) and the uncertainties involved in the model setup were considerable. We concluded that the effect of sea level rise is negligible for salinization in our case.

We agree that the change in salinity can not be seen as a direct result of only the winter precipitation, especially as an increase in precipitation would not be expected to result in an increase in salinity, as the reviewer correctly notes. We would like to change the text in order to reflect these results better: "In those scenarios where there is a positive trend in deep salinity, this seems to be the result of a combination of factors, such as increased $ET$ and changed summer and winter precipitation. A higher $ET$ might affect salinity in two ways: On the one hand, the salt concentration would increase in the remaining soil water when $ET$ increases. On the other hand, the increase in pressure gradient when soils dry out in summer can cause a stronger bottom flux of saline water. In the case of, e. g., scenario IV with the strongest increase in salinity the $ET$ has increased strongly and the summer precipitation decreased. The increase in winter precipitation was apparently not enough to flush the salt from the profiles. While in scenario VI the $ET$ increase was highest, but both summer and winter precipitation increased in this scenario, thus limiting the increase in salinization."

**References**

de Louw, P.G.B., Oude Essink, G.H.P., Stuyfzand, P.J., and van der Zee, S.E.A.T.M.: Upward groundwater flow in boils as the dominant mechanism of salinization in deep polders, The Netherlands, Journal of Hydrology, 394, 494–506, DOI: 10.1016/j.jhydrol.2010.10.009, 2010.

de Louw, P.G.B., van der Velde, Y., and van der Zee, S.E.A.T.M.: Quantifying water and salt fluxes in a lowland polder catchment dominated by boil seepage: a probabilistic end-member mixing approach, Hydrol. Earth Syst. Sci., 15, 2101–2117, DOI: 10.5194/hess-15-2101-2011, 2011.

de Louw, P.G.B., Vandenbohede, A., Werner, A.D., and Oude Essink, G.H.P.: Natural saltwater upconing by preferential groundwater discharge through boils, Journal of Hydrology, 490, 74–87, DOI: 10.1016/j.jhydrol.2013.03.025, 2013.

Doherty, J.: PEST: Model independent parameter estimation, User manual, Watermark Numerical Computing, Brisbane, 2010.

Herbert, E.R., Boon, P., Burgin, A.J., Neubauer, S.C., Franklin, R.B., Ardón, M., Hopfensperger, K.N., Lamers, L.P.M., and Gell, P.: A global perspective on wetland salinization: ecological consequences of a growing threat to freshwater wetlands, Ecosphere 6(10), 206. DOI: 10.1890/ES14-00534.1, 2015.

Kroes, J., Wesseling, J., and van Dam, J.: Integrated modelling of the soil–water–atmosphere–plant system using the model SWAP 2.0 an overview of theory and an application, Hydrological Processes, 14, 1993–2002, 2000.

Oude Essink, G.H.P., van Baaren, E.S., and de Louw, P.G.B.: Effects of climate change on coastal groundwater systems: A modeling study in the Netherlands, Water Resources Research, 46, W00F04, DOI: 10.1029/2009WR008719, 2010.

Wösten, J., Lilly, A., Nemes, A., and le Bas, C.: Development and use of a database of hydraulic properties of European soils, Geoderma, 90, 169–185, 1999.